# Enhancing Batch Normalized Convolutional Networks using Displaced Rectifier Linear Units: A Systematic Comparative Study

## Abstract

In this paper, we turn our attention to the interworking between the activation functions and the batch normalization, which is virtually mandatory technique to train deep networks currently. We propose the activation function Displaced Rectifier Linear Unit (DReLU) by conjecturing that extending the identity function of ReLU to the third quadrant enhances compatibility with batch normalization. Moreover, we used statistical tests to compare the impact of using distinct activation functions (ReLU, LReLU, PReLU, ELU, and DReLU) on the learning speed and test accuracy performance of standardized VGG and Residual Networks state-of-the-art models. These convolutional neural networks were trained on CIFAR-100 and CIFAR-10, the most commonly used deep learning computer vision datasets. The results showed DReLU speeded up learning in all models and datasets. Besides, statistical significant performance assessments ($p < 0.05$) showed DReLU enhanced the test accuracy presented by ReLU in all scenarios. Furthermore, DReLU showed better test accuracy than any other tested activation function in all experiments with one exception, in which case it presented the second best performance. Therefore, this work demonstrates that it is possible to increase performance replacing ReLU by an enhanced activation function.

## 1 Introduction

The recent advances in deep learning research have produced more accurate image, speech, and language recognition systems and generated new state-of-the-art machine learning applications in a broad range of areas such as mathematics, physics, healthcare, genomics, financing, business, agriculture, etc. Although advances have been made, accuracy performance enhancements have usually demanded considerably deeper or more complex models, which tend to increase the required computational resources (processing time and memory usage).

Instead of increasing deep models depth or complexity, a less computational expensive alternative approach to enhance deep learning performance across-the-board is to design more efficient activation functions. Even if computational resources are no issue, to employ enhanced activation functions nevertheless contributes to speeding up learning and achieving higher accuracy.

Indeed, by allowing the training of deep neural networks, the discovery of Rectified Linear Units (ReLU) (Nair & Hinton, 2010; Glorot et al., 2011; Krizhevsky et al., 2012) was one of the main factors that contributed to deep learning advent. ReLU allowed achieving higher accuracy in less time by avoiding the vanishing gradient problem (Hochreiter, 1991). Before ReLU, activation functions such as Sigmoid and Hyperbolic Tangent were unable to train deep neural networks because of the absence of the identity function for positive input.

However, ReLU presents drawbacks. For example, some researchers argument that zero slope avoids learning for negative values (Maas et al., 2013; He et al., 2016b). Therefore, other activation functions like Leaky Rectifier Linear Unit (LReLU) (Maas et al., 2013), Parametric Rectifier Linear Unit (PReLU) (He et al., 2016b) and Exponential Linear Unit (ELU) (Clevert et al., 2015) were proposed (Appendix A). Unfortunately, there is no consensus about how these proposed nonlinearities compare to ReLU, which therefore remains the most used activation function in deep learning.

Similar to activation functions, batch normalization (Ioffe & Szegedy, 2015) currently plays a fundamental role in training deep architectures (Appendix B). This technique normalizes the inputs of each layer, which is equivalent to normalizing the outputs of the deep model previous layer. However, before being used as inputs for the subsequent layer, the normalized data are typically fed into activation functions (nonlinearities), which necessarily skew the otherwise normalized distributions. In fact, ReLU only produces non-negative activations, which is harmful to the previously normalized data. The outputs mean values after ReLU are no longer zero, but rather necessarily positives. Therefore, the ReLU skews the normalized distribution (Section 2).

Aiming to mitigate the mentioned problem, we concentrate our attention on the interaction between activation functions and batch normalization. We conjecture that nonlinearities that are more compatible with batch normalization present higher performance. After that, considering that an identity transformation preserves any statistical distribution, we assume that to extend the identity function from the first quadrant to the third implies less damage to the normalization procedure.

Hence, we investigate and propose the activation function Displaced Rectifier Linear Unit (DReLU), which partially prolongs the identity function beyond origin. Hence, DReLU is essentially a ReLU diagonally displaced into the third quadrant. Different from all other previous mentioned activation functions, the inflection of DReLU does not happen at the origin, but in the third quadrant.

Considering the widespread adoption and practical importance, we used Convolutional Neural Networks (CNN) (LeCun et al., 1998; Krizhevsky et al., 2012) in our experiments. Moreover, as particular examples of CNN architectures, we used the previous ImageNet Large Scale Visual Recognition Competition (ILSVRC) winners Visual Geometry Group (VGG) (Simonyan & Zisserman, 2014) and Residual Networks (ResNets) (He et al., 2016a;c). These architectures have distinctive designs and depth to promote generality to the conclusions of this work. In this regard, we evaluated how replacing the activation function impacts the performance of well established and widely used standard state-of-the-art models. Finally, we decided to employ the two most broadly used computer vision datasets by deep learning research community: CIFAR-100 (Krizhevsky, 2009) and CIFAR-10 (Krizhevsky, 2009).

In this systematic comparative study, performance assessments were carried out using statistical tests with a significance level of 5% (Appendix C.5). At least ten executions of each of experiment were executed. However, when the mentioned significance level was not achieved, ten additional runs were performed.

## 2 DISPLACED RECTIFIER LINEAR UNITS

Consider $x$ the input of a layer composed of a generic transformation $Wx + b$ followed by a nonlinearity, for instance, ReLU. After the addition of the batch normalization layer, the overall joint transformation performed by the block (composed layer) is given by:

$$z = \text{ReLU}(\text{BN}(Wx + b)) \tag{1}$$

For a moment, consider the intermediate activation $y$ produced inside the block:

$$y = \text{BN}(Wx + b) \tag{2}$$

Without loss of generality, assume $\gamma = 1$ and $\beta = 0$ (Ioffe & Szegedy, 2015). Therefore, since $y$ is the output of a batch normalization layer, we can rewrite unbiased estimators for the expected value and variance of any given dimension $k$ as follows (Appendix B):

$$\hat{\mathbb{E}}[\hat{y}^{(k)}] = 0, \widehat{\text{Var}}[\hat{y}^{(k)}] = 1 \tag{3}$$

Consequently, we investigate the expected value and variance of the activations distribution produced by the combined layer. Therefore, if $z$ is the output of the block using a ReLU nonlinearity, it immediately follows that:

$$z = \text{ReLU}(y) \tag{4}$$

The application of ReLU removes all negative values from a distribution. Hence, it necessarily produces positive mean as output. Therefore, it can be written:

$$\hat{\mathbb{E}}[z^{(k)}] = \hat{\mathbb{E}}[\text{ReLU}(\hat{y}^{(k)})] > \hat{\mathbb{E}}[\hat{y}^{(k)}] = 0 \tag{5}$$

Considering that a distribution with $\hat{\mathbb{E}}[\hat{y}^{(k)}] = 0$ has to present negative values, replacing all negative activations with zeros makes the variance of $z^{(k)}$ necessarily lower than the variance of the original distribution $\hat{y}^{(k)}$. Consequently, we can rewrite:

$$\widehat{\text{Var}}[z^{(k)}] = \widehat{\text{Var}}[\text{ReLU}(\hat{y}^{(k)})] < \widehat{\text{Var}}[\hat{y}^{(k)}] = 1 \tag{6}$$

Therefore, despite batch normalization, the activations after the whole block are not perfectly normalized since these outputs present neither zero mean nor unit variance. Consequently, regardless of the presence of a batch normalization layer, after the ReLU, the inputs passed to the next composed layer have neither mean of zero nor variance of one that was the objective in the first place.

In this sense, ReLU skews an otherwise previous normalized output. In other words, ReLU reduces the correction of the internal covariance shift promoted by the batch normalization layer. Consequently, we conclude the ReLU bias shift effect (Clevert et al., 2015) is directly related to the drawback ReLU generates to the batch normalization.

Consequently, we propose DReLU, which is essentially a diagonally displaced ReLU. It generalizes both ReLU and SReLU (Clevert et al., 2015) by allowing its inflection to move diagonally from the origin to any point of the form $(-\delta, -\delta)$. If $\delta = 0$, DReLU becomes ReLU. If $\delta = 1$, DReLU becomes SReLU. Therefore, the slope zero component of the activation function provides negative activations, instead of null ones. Unlike ReLU, in DReLU learning can happen for negative inputs since gradient is not necessarily zero. The following equation defines DReLU:

$$y = \begin{cases} x & \text{if } x \geq -\delta \\ -\delta & \text{if } x < -\delta \end{cases} \tag{7}$$

DReLU can be regarded as a generalization of the Shifted Rectifier Linear Unit (SReLU) (Clevert et al., 2015). In fact, instead of always prolong the identity to the point $(-1, -1)$, in DReLU we established a hyperparameter $\delta$ that defines the most appropriate point $(-\delta, -\delta)$ where the inflection should happen. In this sense, SReLU is a particular case of DReLU where $\delta = 1$. Considering that the experiments we performed to determine the DReLU hyperparameter contemplates $\delta = 1$ as a possible value for $\delta$, it was not necessary to include SReLU in the present comparative study. Indeed, our experiments showed that the addition of the parameter $\delta$ allowed DReLU to significantly outperform SReLU in all of our hyperparameter definition experiments (Appendix C.6). Considering that DReLU replaces ReLU in Eq. 1, the activations of the composed layer become:

$$\boldsymbol{z} = \text{DReLU}(\text{BN}(\boldsymbol{Wx} + \boldsymbol{b})) \tag{8}$$

Since DReLU extends the identity function into the third quadrant, it is no longer possible to conclude Eq. 5 is valid. Therefore, the consequence presented in the mentioned equation is probably at least minimized. In this case, we can conclude that $\hat{\mathbb{E}}[z^{(k)}]_{DReLU}$ is much probably near to zero than $\hat{\mathbb{E}}[z^{(k)}]_{ReLU}$. Hence, we can rewrite:

$$\hat{\mathbb{E}}[z^{(k)}]_{DReLU} < \hat{\mathbb{E}}[z^{(k)}]_{ReLU} \tag{9}$$

Furthermore, DReLU exhibits a noise-robust deactivation state for very negative inputs, a feature not granted by LReLU and PReLU. A noise-robust deactivation state is achieved by setting the slope zero for highly negative values of input (Clevert et al., 2015). Some authors argument that activation functions with this propriety improve learning (Clevert et al., 2015). Finally, DReLU is less computationally complex than LReLU, PReLU, and ELU. In fact, since DReLU has the same shape of ReLU, it essentially has the same computational complexity.

## 3 EXPERIMENTS

In this comparative study, we define an experiment as the training of a deep model using a distinct activation function on a given dataset. If not otherwise mentioned, we conducted ten executions of each experiment. We define a scenario as the set of experiments regarding all activation functions on a specific dataset using a particular model. In this regard, this paper presents the consolidated results of six scenarios (two datasets versus three models) that correspond to 30 experiments, which in turn represents a total of 320 executions (training of deep neural networks). In two cases, we executed 20 instead of 10 runs of a given experiment to achieve the desired statistical significance.

We trained the models during 100 epochs since it was enough to the test accuracy to saturate. At epochs 40 and 70, we evaluated the test accuracy of the partially trained models. Therefore, we were able to assess how fast each model was learning to generalize based on the activation function used by the model. This is important for compare the expected performance of the activation functions in applications where the models need to provide high test accuracy training only a few tens of epochs. Since the training time of an epoch shows no significant difference among the activation functions, we said that the nonlinearity that provided the best test accuracy in these terms to be learning faster. The Appendix C provides a detailed explanation of the performed experiments.

All experiments were conducted without using dropout (Srivastava et al., 2014) since recent studies have shown that, despite improving the training time, dropout provides unclear contributions to the overall deep model performance (Ioffe & Szegedy, 2015). Moreover, dropout has recently become a technique restricted to fully-connected layers, which in turn are being less used and replaced by an average pooling layer in more recent architectures (He et al., 2016a; Huang et al., 2016; He et al., 2016c; Zagoruyko & Komodakis, 2016; 2017). Therefore, since currently fully connected layers are rarely used in modern CNN, the usage of dropout is accordingly becoming unusual. This can be demonstrated by observing that the most recent CNN models are not using dropout, but only batch normalization (He et al., 2016a; Huang et al., 2016; He et al., 2016c; Zagoruyko & Komodakis, 2016; 2017). Particularly in the case of DenseNets (Huang et al., 2016), the results just using batch normalization are significantly better than using both techniques.

This recent tendency of design modern deep networks using only batch normalization but avoiding dropout can also be observed in the discriminative and generative convolutional models recently used in Generative Adversarial Networks (GANs) (Radford et al., 2015). Hence, we emphasize that we designed the experiments of this comparative study to reflect the scenario we believe is currently the most likely and relevant from the perspective of training modern CNNs, which contemplates the use exclusively of batch normalization and no dropout.

The comparative study provided in the paper was designed to be self-contained to avoid misleading comparisons of experiments performed in entirely different situations. In this sense, it should be noticed that the results presented by the papers that proposed the activation functions which are being used in this study (LReLU (Maas et al., 2013), PReLU (He et al., 2016b) and ELU (Clevert et al., 2015)), must not be compared to the ones presented here because of the following reasons.

First, the studies previously mentioned were performed with use of dropout and without batch normalization, which is a technique that was not available when the mentioned studies were conducted. The only exception is ELU, where a few tested scenarios used batch normalization. However, even in those cases, dropout was always and intensively employed. Second, the cited studies did not use standardized models such as VGG or ResNet where the only factor of change was the compared activation functions. For example, in ELU paper, hand designed models were used to compare the performance of the activation function or in some cases completely different models were compared. Third, the results presented by the mentioned works did not use statistical tests. In this sense, considering the variation of the performance of the experiments based on different initialization or data shuffling, the conclusions may not be much trustable from a statistical point of view.

Hence, the conclusions regarding the performance of the cited activation functions based on their original paper may not be valid in the context where only batch normalization, but no dropout, is used to regularize the deep models. In fact, one of the significant contributions of this paper is providing a systematic statistical supported comparative study using standardized models in the currently predominant scenario of using batch normalization without dropout.

## 4 RESULTS AND DISCUSSION

In the following subsections, we analyze the tested scenarios. In each case, we first discuss the activation functions learning speed based on test accuracy obtained for the partially trained models. Subsequently, we comment about the test accuracy performances of the activation functions, which corresponds to the respective model test accuracy evaluated after 100 epochs. Naturally, we consider that an activation function presents better test accuracy if it showed the higher test accuracy for the final trained models on a particular dataset.

In all scenarios, the null hypotheses were the test accuracy samples taken from different activation functions originated from the same distribution. In other works, all the compared activation functions have the same test accuracy performance in the particular scenario. The null hypotheses were rejected for all scenarios (Table 1), which means that with statistical significance ($p < 0.05$) at least one of the activation functions presents a test accuracy performance that is different from the others activation functions. Therefore, we used the Conover-Iman post-hoc tests for pairwise multiple comparisons for all combination of datasets and models (Tables 3, 4, 5, 7, 8, 9). In these tables, the best results and $p$-values of the comparison of DReLU to other activation functions are in bold.

Table 1: Kruskal-Wallis test results

| | CIFAR-100 | | | CIFAR-10 | | |
|---|---|---|---|---|---|---|
| Score | VGG-19 | ResNet-56 | ResNet-110 | VGG-19 | ResNet-56 | ResNet-110 |
| $\tilde{\chi}^2(4)$ | 44.918 | 41.253 | 41.169 | 56.087 | 40.115 | 49.451 |
| $p$-value | $4.135 \times 10^{-9}$ | $2.383 \times 10^{-8}$ | $2.48 \times 10^{-8}$ | $1.922 \times 10^{-11}$ | $4.097 \times 10^{-8}$ | $4.7 \times 10^{-10}$ |

### 4.1 CIFAR-100 DATASET

The Table 2 presents the mean of the nonlinearities layers mean activations performed in the CIFAR-100 training dataset. It shows that DReLU is more capable of reducing the bias shift effect during training than ReLU. Therefore, as expected and in agreement with 9, all of our experiments showed that the identity mapping extension produced less damage to the normalization performed by the previous layer and in fact mitigated the bias shift effect when compared to ReLU.

Table 2: CIFAR-100 averaged nonlinearities layers mean activations

| | 40 Epochs Performance Evaluation | | |
|---|---|---|---|
| Nonlinearity | VGG-19 | ResNet-56 | ResNet-110 |
| ReLU | $0.0950 \pm 0.0016$ | $0.1288 \pm 0.0016$ | $0.0852 \pm 0.0020$ |
| DReLU | $0.0578 \pm 0.0010$ | $0.0972 \pm 0.0030$ | $0.0551 \pm 0.0012$ |

| | 70 Epochs Performance Evaluation | | |
|---|---|---|---|
| Nonlinearity | VGG-19 | ResNet-56 | ResNet-110 |
| ReLU | $0.0744 \pm 0.0008$ | $0.1058 \pm 0.0015$ | $0.0680 \pm 0.0015$ |
| DReLU | $0.0362 \pm 0.0009$ | $0.0743 \pm 0.0024$ | $0.0390 \pm 0.0008$ |

| | 100 Epochs Performance Evaluation | | |
|---|---|---|---|
| Nonlinearity | VGG-19 | ResNet-56 | ResNet-110 |
| ReLU | $0.0640 \pm 0.0006$ | $0.0943 \pm 0.0013$ | $0.0594 \pm 0.0013$ |
| DReLU | $0.0261 \pm 0.0007$ | $0.0618 \pm 0.0023$ | $0.0300 \pm 0.0007$ |

In relation to VGG-19, the experiments showed DReLU outperformed the test accuracy results of ReLU and all other assessed activation functions on either 40 and 70 epochs. In this sense, the results demonstrated that DReLU produced the fastest training (Fig. 1). Moreover, DReLU presented the best final test performance (Table 3) in this case.

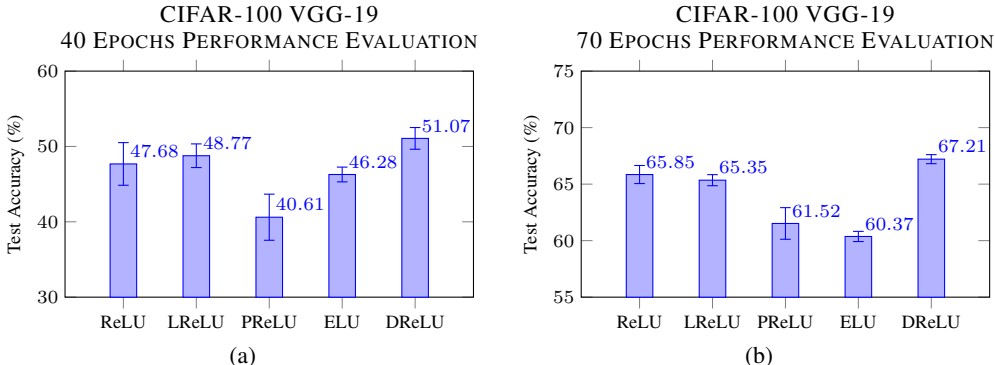

Figure 1: VGG-19 model test accuracy means and standard deviations on the CIFAR-100 dataset. (a) 40 trained epochs. (b) 70 trained epochs.

Table 3: CIFAR-100 VGG-19 100 epochs performance evaluation
Test accuracy means, standard deviations and post hoc tests $p$-values

| Unit | Accuracy (%) | ReLU | LReLU | PReLU | ELU |
|------|--------------|------|-------|-------|-----|
| ReLU | $73.40 \pm 0.22$ | - | - | - | - |
| LReLU | $73.01 \pm 0.22$ | $p < 2.2 \times 10^{-5}$ | - | - | - |
| PReLU | $71.55 \pm 0.24$ | $p < 1.7 \times 10^{-13}$ | $p < 1.1 \times 10^{-6}$ | - | - |
| ELU | $67.57 \pm 0.17$ | $p < 2 \times 10^{-16}$ | $p < 5.6 \times 10^{-14}$ | $p < 6.7 \times 10^{-6}$ | - |
| **DReLU** | $\mathbf{73.69 \pm 0.21}$ | $\mathbf{p < 0.00012}$ | $\mathbf{p < 1.5 \times 10^{-11}}$ | $\mathbf{p < 2 \times 10^{-16}}$ | $\mathbf{p < 2 \times 10^{-16}}$ |

In the case of ResNet-56, DReLU overcame the test accuracy results of the other activation functions on either 40 and 70 epochs once again. Therefore, we concluded DReLU generated the fastest learning (Fig. 2). Regarding test accuracy, DReLU outperformed ReLU ($p < 0.00228$) and all other options, with exception to LReLU. Although, no statistical significance was achieved in this pairwise comparison (Table 4).

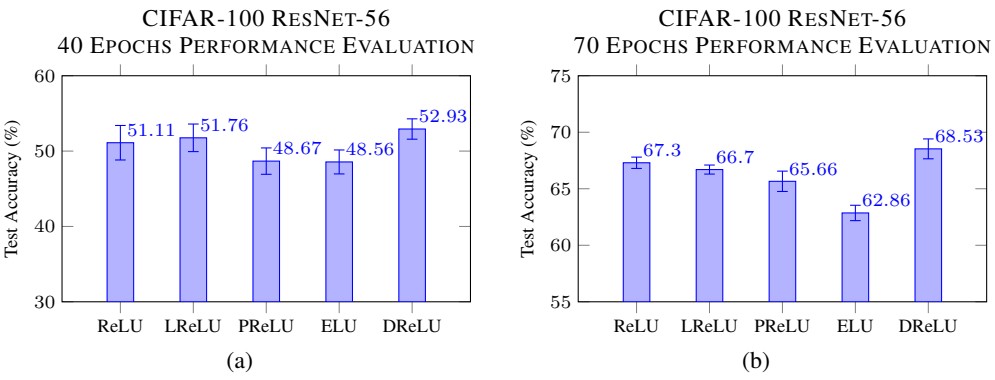

Figure 2: ResNet-56 model test accuracy means and standard deviations on the CIFAR-100 dataset. (a) 40 trained epochs. (b) 70 trained epochs.

Table 4: CIFAR-100 ResNet-56 100 epochs performance evaluation
Test accuracy means, standard deviations and post hoc tests $p$-values

| Unit | Accuracy (%) | ReLU | LReLU | PReLU | ELU |
|------|-------------|------|-------|-------|-----|
| ReLU | $73.70 \pm 0.22$ | - | - | - | - |
| LReLU | $\mathbf{74.20 \pm 0.31}$ | $p < 6.2 \times 10^{-5}$ | - | - | - |
| PReLU | $72.53 \pm 0.29$ | $p < 1.5 \times 10^{-5}$ | $p < 5.4 \times 10^{-12}$ | - | - |
| ELU | $70.13 \pm 0.41$ | $p < 5.7 \times 10^{-11}$ | $p < 2 \times 10^{-16}$ | $p < 0.00059$ | - |
| DReLU | $\mathbf{74.12 \pm 0.38}$ | $\mathbf{p < 0.00228}$ | $\mathbf{p < 0.24300}$ | $\mathbf{p < 2.7 \times 10^{-10}}$ | $\mathbf{p < 2.4 \times 10^{-15}}$ |

In ResNet-100, DReLU also provided the fastest learning in all situations (Fig. 3). Finally, DReLU test accuracy outperformed ReLU ($p < 2.3 \times 10^{-5}$) and all others activation functions again (Table 5). Therefore, in CIFAR-100 as a whole, DReLU presented the fastest learning for all three models considered. The results showed DReLU always outperformed ReLU test accuracy in all studied models. Besides, DReLU was the most accurate in two evaluated models and the second in the other scenario.

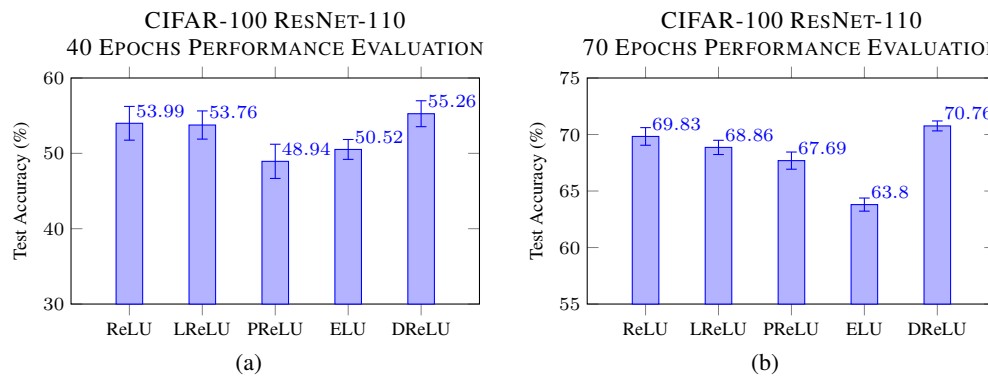

Figure 3: ResNet-110 model test accuracy means and standard deviations on the CIFAR-100 dataset. (a) 40 trained epochs. (b) 70 trained epochs.

Table 5: CIFAR-100 ResNet-110 100 epochs performance evaluation
Test accuracy means, standard deviations and post hoc tests $p$-values

| Unit | Accuracy (%) | ReLU | LReLU | PReLU | ELU |
|------|-------------|------|-------|-------|-----|
| ReLU | $75.70 \pm 0.28$ | - | - | - | - |
| LReLU | $75.94 \pm 0.29$ | $p < 0.0286$ | - | - | - |
| PReLU | $74.86 \pm 0.37$ | $p < 1.2 \times 10^{-5}$ | $p < 5.3 \times 10^{-9}$ | - | - |
| ELU | $70.90 \pm 0.42$ | $p < 3.6 \times 10^{-11}$ | $p < 2.9 \times 10^{-14}$ | $p < 0.0005$ | - |
| DReLU | $\mathbf{76.20 \pm 0.31}$ | $\mathbf{p < 2.3 \times 10^{-5}}$ | $\mathbf{p < 0.0286}$ | $\mathbf{p < 1.2 \times 10^{-5}}$ | $\mathbf{p < 3.6 \times 10^{-11}}$ |

## 4.2 CIFAR-10 DATASET

The Table 6 presents the averaged nonlinearities layers mean activations performed in the CIFAR-10 training dataset. It shows again that DReLU is more efficient to reduce the bias shift effect during training than ReLU. Hence, as suggested by 9, the experiments showed that the identity mapping extension produced less damage to the normalization performed by the previous layer and indeed mitigated the bias shift effect when compared to ReLU.

Table 6: CIFAR-10 averaged nonlinearities layers mean activations

| Nonlinearity | 40 Epochs Performance Evaluation | | |
| --- | --- | --- | --- |
| | VGG-19 | ResNet-56 | ResNet-110 |
| ReLU | $0.0630 \pm 0.0009$ | $0.1006 \pm 0.0027$ | $0.0669 \pm 0.0009$ |
| DReLU | $0.0293 \pm 0.0010$ | $0.0671 \pm 0.0016$ | $0.0376 \pm 0.0011$ |

| Nonlinearity | 70 Epochs Performance Evaluation | | |
| --- | --- | --- | --- |
| | VGG-19 | ResNet-56 | ResNet-110 |
| ReLU | $0.0449 \pm 0.0006$ | $0.0783 \pm 0.0014$ | $0.0515 \pm 0.0006$ |
| DReLU | $0.0139 \pm 0.0003$ | $0.0471 \pm 0.0011$ | $0.0244 \pm 0.0009$ |

| Nonlinearity | 100 Epochs Performance Evaluation | | |
| --- | --- | --- | --- |
| | VGG-19 | ResNet-56 | ResNet-110 |
| ReLU | $0.0364 \pm 0.0004$ | $0.0673 \pm 0.0011$ | $0.0435 \pm 0.0005$ |
| DReLU | $0.0066 \pm 0.0002$ | $0.0351 \pm 0.0008$ | $0.0161 \pm 0.0007$ |

In VGG-19 model, DReLU provided faster learning than any other activation function for either 40 and 70 epochs (Fig. 4). Moreover, DReLU presented the best test accuracy performance (Table 7). We performed 20 executions of either DReLU and ReLU experiment.

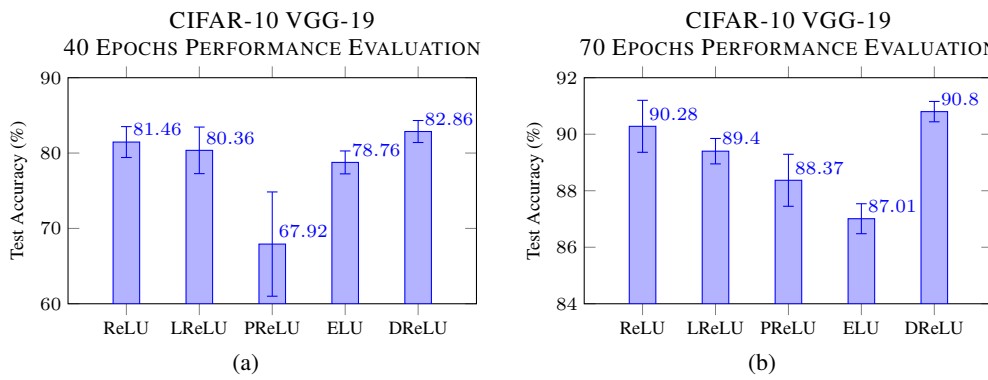

Figure 4: VGG-19 model test accuracy means and standard deviations on the CIFAR-10 dataset. (a) 40 trained epochs. (b) 70 trained epochs.

Table 7: CIFAR-10 VGG-19 100 epochs performance evaluation
Test accuracy means, standard deviations and post hoc tests $p$-values

| Unit | Accuracy (%) | ReLU | LReLU | PReLU | ELU |
| --- | --- | --- | --- | --- | --- |
| ReLU | $93.82 \pm 0.13$ | - | - | - | - |
| LReLU | $93.35 \pm 0.18$ | $p < 1.4 \times 10^{-9}$ | - | - | - |
| PReLU | $93.27 \pm 0.22$ | $p < 2.2 \times 10^{-10}$ | $p < 0.694\,49$ | - | - |
| ELU | $91.40 \pm 0.15$ | $p < 2 \times 10^{-16}$ | $p < 0.000\,22$ | $p < 0.000\,81$ | - |
| DReLU | $\mathbf{93.92 \pm 0.11}$ | $\mathbf{p < 0.00330}$ | $\mathbf{p < 5.6 \times 10^{-14}}$ | $\mathbf{p < 9.1 \times 10^{-15}}$ | $\mathbf{p < 2 \times 10^{-16}}$ |

In the case of ResNet-56, DReLU provided faster learning than any other nonlinearity on either 40 and 70 epochs in this scenario (Fig. 5). Furthermore, DReLU was again the most accurate followed by ReLU (Table 8).

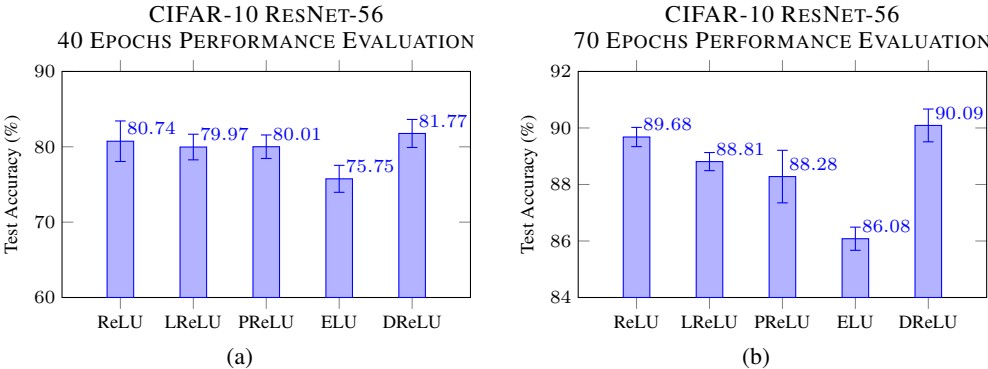

Figure 5: ResNet-56 model test accuracy means and standard deviations on the CIFAR-10 dataset. (a) 40 trained epochs. (b) 70 trained epochs.

Table 8: CIFAR-10 ResNet-56 100 epochs performance evaluation
Test accuracy means, standard deviations and post hoc tests $p$-values

| Unit | Accuracy (%) | ReLU | LReLU | PReLU | ELU |
|------|--------------|------|-------|-------|-----|
| ReLU | $93.29 \pm 0.21$ | - | - | - | - |
| LReLU | $93.01 \pm 0.21$ | $p < 1.2 \times 10^{-6}$ | - | - | - |
| PReLU | $93.01 \pm 0.10$ | $p < 4.7 \times 10^{-5}$ | $p < 0.9726$ | - | - |
| ELU | $90.21 \pm 0.11$ | $p < 4.0 \times 10^{-13}$ | $p < 1.2 \times 10^{-6}$ | $p < 1.3 \times 10^{-6}$ | - |
| **DReLU** | $\mathbf{93.52 \pm 0.14}$ | $\boldsymbol{p < 0.0027}$ | $\boldsymbol{p < 1.1 \times 10^{-9}}$ | $\boldsymbol{p < 1.0 \times 10^{-9}}$ | $\boldsymbol{p < 2 \times 10^{-16}}$ |

In relation to ResNet-110, DReLU provided the fastest leaning on either 40 and 70 epochs once more (Fig. 6). DReLU was the most accurate solution also for this scenario (Table 9). In this case, we also performed 20 runs of DReLU and ReLU.

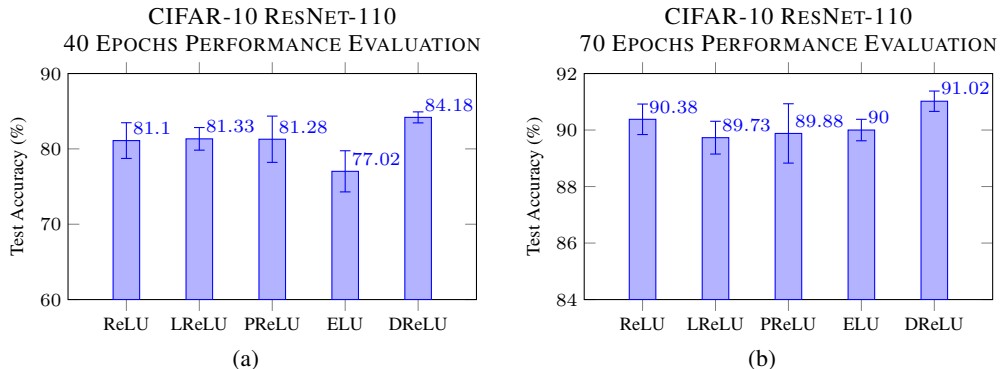

Figure 6: ResNet-110 model test accuracy means and standard deviations on the CIFAR-10 dataset. (a) 40 trained epochs. (b) 70 trained epochs.

Hence, in the CIFAR-10 as a whole, DReLU also presented the best learning speed for all considered models. Moreover, the results showed DReLU again surpassed ReLU test accuracy in all analyzed scenarios. Furthermore, DReLU was the most accurate activation function in all evaluated models.

Table 9: CIFAR-10 ResNet-110 100 epochs performance evaluation
Test accuracy means, standard deviations and post hoc tests $p$-values

| Unit | Accuracy (%) | ReLU | LReLU | PReLU | ELU |
|------|--------------|------|-------|-------|-----|
| ReLU | $94.00 \pm 0.18$ | - | - | - | - |
| LReLU | $93.56 \pm 0.26$ | $p < 1.6 \times 10^{-7}$ | - | - | - |
| PReLU | $93.74 \pm 0.09$ | $p < 2.3 \times 10^{-5}$ | $p < 0.25754$ | - | - |
| ELU | $90.76 \pm 0.29$ | $p < 1.8 \times 10^{-13}$ | $p < 0.00473$ | $p < 0.00013$ | - |
| **DReLU** | $\mathbf{94.11 \pm 0.18}$ | $\mathbf{p < 0.03614}$ | $\mathbf{p < 1.6 \times 10^{-7}}$ | $\mathbf{p < 2.3 \times 10^{-5}}$ | $\mathbf{p < 1.8 \times 10^{-13}}$ |

## 4.3 DISCUSSION

Primarily, we reemphasize the studies that proposed the activation functions compared in this paper used significantly different (specifically designed) models from the (standardized) ones used in this study. In this sense, it is not possible to make a direct comparison between their results and the ones presented in this work. In fact, in the mentioned papers, the usage regular of dropout may have produced a no ideal performance from ReLU since its fast training capacity was probably reduced. In the mentioned papers, no experiments were executed using batch normalization without dropout. As this study presents a significantly different scenario, we can expect different conclusions from this work. Moreover, this paper performed statistical tests to prove that, in the conditions which experiments were executed, the proposed solution is consistently better than the other options. To do that, we performed a significant number of executions of each experiment. It is an important point to consider since the performance of same models presents a slight variation each time they are trained on a given dataset.

Taking into consideration the previous comments, in our experiments, regarding the CIFAR-100 dataset, DReLU presented the fastest learning for all three models considered. Moreover, the results showed DReLU always outperformed ReLU test accuracy in all studied models. Besides, DReLU was the most accurate in two evaluated models and the second in the other one. In the CIFAR-10 dataset, DReLU also presented the best learning speed for all considered models. Moreover, the results showed DReLU surpassed ReLU test accuracy in all analyzed scenarios. Actually, DReLU was the most accurate activation function in all evaluated models.

It is important to mention that we commonly observed that ReLU usually produced the second best training speed and test accuracy performance. This apparent surprise result may be explained by the use of batch normalization. Indeed, the correction of the internal covariate shift problem enabled by the batch normalization technique acted relatively in benefit of ReLU and detriment of the other previously proposed units. Hence, batch normalization significantly helped to avoid the so-called "dying ReLU" problem (Karpathy, 2017; Maas et al., 2013; Cunningham et al., 2017).

In fact, even if a substantial gradient pushes the weights to the zero gradient region of ReLU, the normalization process tends to bring them back to inflection region of ReLU, which avoids the ReLU to die. This fact can explain why ReLU typically outperforms LReLU, PReLU, and ELU in these situations but apparently did not when these activation functions were proposed a few years ago before the batch normalization advent.

The fact that batch normalization relatively helped ReLU in detriment of LReLU, PReLU and ELU make particularly impressive the ability of DReLU to overcome the performance of ReLU in exclusively batch normalized networks. Particularly remarkable is the ability of DReLU to enhance the training speed during the first decades significantly.

## 5 CONCLUSION

In this paper, we have proposed a novel activation function for deep learning architectures, referred to as DReLU. The results showed that DReLU presented better learning speed than the all alternative activation functions, including ReLU, in all models and datasets. Moreover, the experiments showed DReLU was more accurate than ReLU in all situations. Besides, DReLU also outperformed test accuracy results of all others investigated activation functions (LReLU, PReLU, and ELU) in all scenarios with one exception. The experiments used batch normalization but avoided dropout. Furthermore, they were designed to cover standard and commonly used datasets (CIFAR-100 and CIFAR-10) and models (VGG and Residual Networks) of several depths and architectures.

In addition to enhancing deep learning performance (learning speed and test accuracy), DReLU is less computationally expensive than LReLU, PReLU, and ELU. Moreover, the mentioned gains were obtained by just replacing the activation function of the model, without any increment in depth or architecture complexity, which usually increases the computational resource requirements as processing time and memory usage.

This paper showed that the batch normalization procedure acted in the benefice of ReLU while other previews proposed activation functions appear not to perform as expected. We believe this happened because batch normalization avoids the so-called "dying ReLU" problem, something that others activation functions were already not affected by in first place.

Furthermore, considering some evaluated models included skip connections, which are a tendency in the design of deep architectures like ResNets, we conjecture the results may generalize to other deep architectures such DenseNets (Huang et al., 2016) that also use this structure.

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

# A  ACTIVATION FUNCTIONS

Currently, all major activation functions adopt the identity transformation to positive inputs, some particular function for negative inputs, and an inflection on the origin. In the following subsections, we describe the activation functions compared in this work.

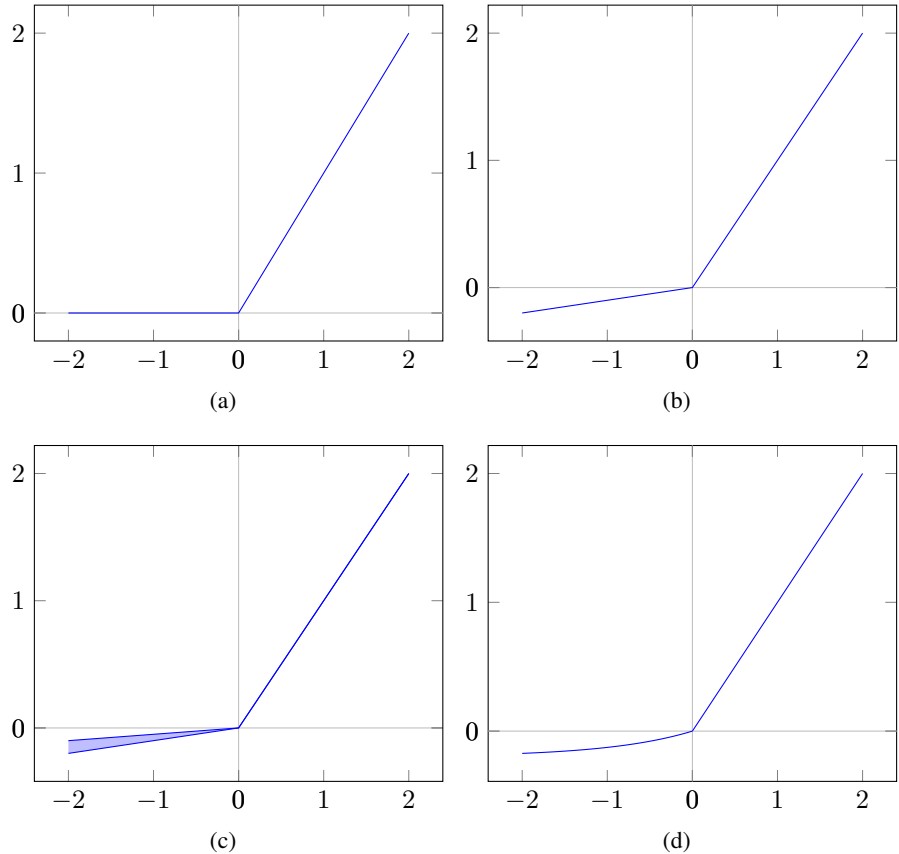

Figure 7: Previous proposed activation function. (a) ReLU. (b) LReLU. (c) PReLU. (d) ELU.

## A.1  RELU

ReLU has become the standard activation function used in deep networks (Fig. 7a). Its simplicity and high performance are the main factors behind this fact. The follow equation defines ReLU:

$$y = \begin{cases} x & \text{if } x \geq 0 \\ 0 & \text{if } x < 0 \end{cases} \tag{10}$$

The Eq. (10) implies ReLU has slope zero for negative inputs and slope one for positive values. It was first used to improve the performance of Restricted Boltzmann Machines (RBM) (Nair & Hinton, 2010). After that, ReLU was used in other neural networks architectures (Glorot et al., 2011). Finally, ReLU has provided superior performance in the supervised training of convolutional neural network models (Krizhevsky et al., 2012).

The identity for positive input values produced high performance by avoiding the vanishing gradient problem (Hochreiter, 1991). A conceivable drawback is the fact ReLU necessarily generates positive mean outputs or activations, which generate the bias shift effect (Clevert et al., 2015). Consequently, while the identity for positive inputs is unanimously accepted as a reliable design option, there is no consensus on how to define promising approaches for negative values.

## A.2  LRELU

LReLU was introduced during the study of neural network acoustic models (Maas et al., 2013) (Fig. 7b). This activation function was proposed to avoid slope zero for negative inputs. The following equation defines LReLU:

$$y = \begin{cases} x & \text{if } x \geq 0 \\ \beta x & \text{if } x < 0 \end{cases} \tag{11}$$

LReLU has no zero slope if $\beta \neq 0$. In fact, it was designed to allow learning to happen even for negative inputs (He et al., 2016b). Moreover, since LReLU does not necessarily produce only positive activations, the bias shift effect may be reduced.

## A.3  PRELU

PReLU is also defined by the Eq. 11, but in this case $\beta$ is a learnable rather than a fixed parameter (He et al., 2016b) (Fig. 7c). The idea behind the PReLU design is to learn the best slope for negative inputs. However, this approach may implicate in overfitting since the learnable parameters may adjust to specific characteristics of the training data.

## A.4  ELU

ELU has inspiration in the natural gradient (Amari, 1998; Clevert et al., 2015) (Fig. 7d). Similarly to LReLU and PReLU, ELU avoids producing necessarily positive mean outputs by allowing negative activation for negative inputs. The Eq. 12 defines ELU:

$$y = \begin{cases} x & \text{if } x \geq 0 \\ \alpha(exp(x) - 1) & \text{if } x < 0 \end{cases} \tag{12}$$

The main drawback of ELU is its higher computational complexity when compared to activation functions such as ReLU, LReLU, and DReLU.

# B    BATCH NORMALIZATION

In machine learning, normalizing the distribution of the input data decreases the training time and improves test accuracy (Tax & Duin, 2002). Consequently, normalization also improves neural networks performance (LeCun et al., 2012). A standard approach to normalizing input data distributions is the mean-standard technique. In this method, the input data is transformed to present zero mean and standard deviation of one.

However, if instead of working with shallow machine learning models, we are dealing with deep architectures; the problem becomes more sophisticated. Indeed, in a deep structure, the output of a layer works as input data to the next. Therefore, in this sense, each layer of a deep model has his own "input data" that is composed of the previous layer output or activations. The only exception is the first layer, for which the input is the original data.

In fact, consider that the stochastic gradient decent (SGD) is being used to optimize the parameters $\theta$ of a deep model. Assume $S$ a sample of $m$ training examples in a mini-batch; then the SGD minimizes the loss given by the equation:

$$\Theta = \arg\min_{\Theta} \frac{1}{m} \sum_{S} L(S, \Theta) \tag{13}$$

Furthermore, consider $x$ the output of the layer $i-1$. These activations are fed into layer $i$ as inputs. In turn, the outputs of layer $i$ are fed into layer $i+1$ producing the overall loss given bellow:

$$L = G(F_{i+1}(F_i(x, \Theta_i), \Theta_{i+1})) \tag{14}$$

In the above equation, $F_{i+1}$ and $F_i$ are the transformation produced by the layers $i+1$ and $i$, respectively. The $G$ function represents the mapping perpetrated by the above layers combined with the loss function adopted as the criterion. Considering that the output of layer $i$ is given by $y = F_i(x, \Theta_i)$, we can rewrite the above equation as follows:

$$L = G(F_{i+1}(y, \Theta_{i+1})) \tag{15}$$

Applying equation (13) to equation (15) and considering a learning rate $\lambda$, we can write the equation to update the parameters of the layer $i+1$ as the follows:

$$\Theta_{i+1} \leftarrow \Theta_{i+1} - \frac{\lambda}{m} \sum_{y} \frac{\partial G(F_{i+1}(y, \Theta_{i+1}))}{\partial \Theta_{i+1}} \tag{16}$$

In fact, the above equation is mathematically equivalent to training the layer $i+1$ on the input data given by $y$, which in turn is the output of the previous layer. Therefore, indeed, we can understand the output of the previous layer as "input data" for the effect of training the current layer using SGD.

Considering each layer has its own "input data" (the output of the previous layer), normalizing only the actual input data of a deep neural network produces a limited effect in enhancing learning speed and test accuracy. Moreover, during the training process, the distribution of the input of each layer changes, which makes training even harder. Indeed, the parameters of a layer are updated while its input (the activations of the previous layer) is modified.

This phenomenon is called internal covariant shift, which is a major factor that hardens the training of deep architectures (Ioffe & Szegedy, 2015). In fact, while the data of shallow models is normalized and static during training, the input of a deep model layer, which is the output of the previous one, is neither a priori normalized nor static throughout training.

Batch normalization is an effective method to mitigate the internal covariant shift (Ioffe & Szegedy, 2015). This approach, which significantly improves training speed and test accuracy, proposes normalizing the inputs of the layers when training deep architectures.

The layers inputs normalization is performed after the each mini-batch training to synchronizing with the deep network parameters update. Therefore, when using batch normalization, for a input $\boldsymbol{x} = (x^{(1)}, x^{(2)}, \ldots, x^{(d)})$, each individual dimension is transformed as follows:

$$\hat{x}^{(k)} = \frac{x^{(k)} - \hat{\mathbb{E}}[x^{(k)}]}{\sqrt{\widehat{\mathbb{Var}}[x^{(k)}]}} \tag{17}$$

## C    EXPERIMENTS DETAILS

The experiments were executed on a machine configured with an Intel(R) Core(TM) i7-4790K CPU, 16 GB RAM, 2 TB HD and a GeForce GTX 980Ti card. The operational system was Ubuntu 14.04 LTS with CUDA 7.5, cuDNN 5.0, and Torch 7 deep learning library.

### C.1    DATASETS, PREPROCESSING AND DATA AUGMENTATION

We trained the models on the CIFAR-100 and CIFAR-10 datasets. The CIFAR-100 image dataset aggregates 100 classes, which in turn contain 600 example images each. From these, 500 are used for training, and 100 are employed for testing. Each example is a 32x32 RGB image. The CIFAR-10 dataset possesses ten classes containing 6000 images, from which 5000 are used for training, and 1000 are left for testing. Again, each training example is an RGB image of size 32x32.

The experiments used regular mean-standard preprocessing. Therefore, each feature was normalized to present zero mean and unit standard deviation throughout that training data. The features were also redimensioned using the same parameters before performing the inference during test. The data augmentation performed was randomized horizontal flips and random crops. Therefore, before training, each image was flipped horizontally with a 0.5 probability. Moreover, four pixels reflected from the picture opposite sides were added to expand it vertically and horizontally. Finally, a 32x32 random crop was taken from the enlarged image. The random crop was then used to train.

### C.2    ACTIVATION FUNCTIONS PARAMETRIZATION

In this paper, we used the parameters originally proposed by the activation function designers (He et al., 2016b; Clevert et al., 2015) since these have been kept in subsequent papers (Shah et al., 2016; Heusel et al., 2015). Therefore, we kept $\beta = 0.25$ for both LReLU and PReLU, and for ELU we maintained $\alpha = 1.0$. We decided to keep those parameters because we consider the original authors and the follower papers, which respectively proposed and kept the original parameterizations, performed hyperparameter search and validation procedures to estimate parameter values that provide high performance for their proposed or used work.

### C.3    MODELS AND INITIALIZATION

As a particular instance of a VGG model, we used the VGG variant with nineteen layers (VGG-19). To train models with a considerably different number of layers, we chose the pre-activation ResNet with fifty-six layers (ResNet-56) and also the pre-activation ResNet with one hundred ten layers (ResNet-110). We employed pre-activation ResNets because they present better performance when compared to the original aproach (He et al., 2016a). The experiments used the Kaiming initialization (He et al., 2016b).

### C.4    TRAINING AND REGULARIZATION

The experiments used an initial learning rate of 0.1, and a learning rate decay of 0.2 with steps in the epochs 60, 80 and 90 for both CIFAR-10 and CIFAR-100 datasets. The experiments employed mini-batches of size 128 and stochastic gradient descent with Nesterov acceleration technique as the optimization method. The moment was set to 0.9, and the weight decay was equal to 0.0005.

### C.5    PERFORMANCE ASSESSMENT

To make assessments about the performance of activation functions, we chose the Kruskal-Wallis one-way analysis of variance statistical tests (Kruskal & Wallis, 1952) because the weight initialization was different for each experiment execution. Consequently, we had independent and also possibly different size samples for the test accuracy distributions. Moreover, the Kruskal-Wallis tests can also be used to confront samples obtained from more than two sources at once, which is appropriated in our study since, for a given scenario, we are comparing five activation functions simultaneously. Besides, it does not assume a normal distribution of the residuals, which makes it more general than the parametric equivalent one-way analysis of variance (ANOVA). We used the Conover-Iman post-hoc tests (Conover & Iman, 1979) for pairwise multiple comparisons.

## C.6 HYPERPARAMETER EXPERIMENTS

Clearly, $\delta$ must not go to infinity because we would lose the nonlinearity in such a case. Therefore, a compromise that causes less damage to the normalization while keeps the activation function nonlinearity has to be achieved. To define the hyperparameter value of DReLU, we performed the experiments showed in Table 10. The results are the mean and standard deviation values of each experiment five executions. Based on these experimental results, we decided to set $\delta = 0.05$. The blue line presents the performance of SReLU, which can be observed is considerably lower than the DReLU using the chosen hyperparameter $\delta = 0.05$.

Table 10: Hyperparameter experiments
Displaced Rectifier Linear Unit
CIFAR-10 test accuracy

| $\delta$ | VGG-19 (%) | ResNet-56 (%) | ResNet-110 (%) | Mean (%) |
|---|---|---|---|---|
| 0.01 | 93.89±0.20 | 93.45±0.13 | 93.98±0.19 | 93.77 |
| 0.05 | 93.91±0.18 | **93.50±0.07** | **94.10±0.10** | **93.83** |
| 0.10 | **94.00±0.13** | 93.43±0.25 | 93.93±0.14 | 93.78 |
| 0.25 | 93.67±0.19 | 92.93±0.19 | 93.31±0.12 | 93.30 |
| 0.50 | 92.70±0.08 | 91.62±0.37 | 92.04±0.17 | 92.12 |
| 1.00 | 90.96±0.16 | 88.64±0.33 | 89.26±0.42 | 89.62 |

