# OpenReview forum: "Enhancing Batch Normalized Convolutional Networks using Displaced Rectifier Linear Units: A Systematic Comparative Study"
_ICLR.cc/2018/Conference — Reject_

### Official Review · AnonReviewer2 · 2017-11-27
**Compare against usage of BN after ReLU**

**Rating:** 3
**Confidence:** 5

**Review:**

The key argument authors present against ReLU+BN is the fact that using ReLU after BN skews the values resulting in non-normalized activations. Although the BN paper suggests using BN before non-linearity many articles have been using BN after non-linearity which then gives normalized activations (https://github.com/ducha-aiki/caffenet-benchmark/blob/master/batchnorm.md) and also better overall performance. The approach of using BN after non-linearity is termed "standardization layer" (https://arxiv.org/pdf/1301.4083.pdf). I encourage the authors to validate their claims against simple approach of using BN after non-linearity.

---

> ### Author Response · Authors · 2017-12-04
> **Answers to comments of Reviewer**
>
> The primary aim of this paper is to propose an activation function to improve the performance of mainstream state-of-the-art convolutional neural networks.
>
> Therefore, the experiments were designed to use the batch normalization followed by ReLU (BN+ReLU) since we believe this is currently clearly the mainstream approach used by most recent proposed state-of-the-art models.
>
> Firstly, the original batch normalization paper, as the reviewer acknowledges, proposed BN+ReLU instead of ReLU+BN. The authors made their arguments why not to use BN+ReLU.
>
> Further, the mentioned approach was followed by the ILSVRC winner ResNet in both the original as much as in the pre-activation variant. To the best of our knowledge, all Generative Adversarial Networks (GANs) use normalization before non-linearity in either Generator and Discriminator convolutional networks. The same is true for (Variational or not) Autoencoder designs.
>
> Besides, the same pattern was observed by the so-called Wide Residual Networks, which showed improved results in some situations compared with original Residual Networks. Furthermore, all the Google's Inception Networks variants from version two to version four followed the same design, placing the non-linearity after the batch normalization.
>
> Finally, this year, DenseNets, which won the CVPR 2017 Best Paper Awards, also insisted on using BN+ReLU, not otherwise. Hence, it is still to be seen if relevant peer-review papers conclude ReLU+BN provides any improvement. Even if this hypothesis proves to be right in future, this will not invalidate the conclusions of the present work, rather it will be a novel achievement not directly related to the present research.
>
> Naturally, it is possible that shortly using ReLU before BN shows better results than otherwise, but it is yet to be demonstrated as none of the most recent and distinguished models did not adopt the mentioned approach. Consequently, we believe much more evidence is still needed to conclude otherwise.
>
> Finally, we believe that the theoretical and mathematical arguments we made still holds. Since DReLU extends the linearity into the third quadrant, we think DReLU+BN is likely to work better than ReLU+BN.

---

### Official Review · AnonReviewer3 · 2017-11-27
**Simple idea, Need comparison in an uncontrolled setting**

**Rating:** 4
**Confidence:** 4

**Review:**

This paper proposes an activation function, called displaced ReLU (DReLU), to improve the performance of CNNs that use batch normalization. Compared to ReLU, DReLU cut the identity function at a negative value rather than the zero. As a result, the activations outputted by DReLU can have a mean closer to 0 and a variance closer to 1 than the standard ReLU. The DReLU is supposed to remedy the problem of covariate shift better.

The presentation of the paper is clear. The proposed method shows encouraging results in a controlled setting (i.e., all other units, like dropout, are removed). Statistical tests are performed for many of the experimental results, which is solid.

However, I have some concerns.
1) As DReLU(x) = max{-\delta, x}, what is the optimal strategy to determine \delta? If it is done by hyperparameter tuning with cross-validation, the training cost may be too high.
2) I believe the control experiments are encouraging, but I do not agree that other techniques like Dropouts are not useful. Using DReLU to improve the state-of-art neural network in an uncontrolled setting is important. The arguments for skipping this experiments are respectful, but not convincing enough.
3) Batch normalization is popular, especially for the convolutional neural networks. However, its application is not universal, which can limit the use of the proposed DReLU. It is a minor concern, anyway.

---

> ### Author Response · Authors · 2017-12-04
> **Answers to comments of Reviewer**
>
> Regarding the second and third comments, we emphasize that the work proposes to design a non-linearity that can be used to improve the performance (training speed and test accuracy) of mainstream state-of-the-art (SOTA) convolutional models.
>
> We are not claiming dropout is useless in general. It is undoubtedly important and frequently used in, for example, Recurrent Neural Networks. However, we firmly believe that its usage in mainstream state-of-the-art convolutional models has been in visible decline recently.
>
> Therefore, designing nonlinearities that can overcome ReLU using dropout (as we believe it may be the case of previously proposed activation functions) would be of no practical significance (for the scope we are considering) if the overall best performance is still achieved by a strictly batch normalized network using ReLU.
>
> In this sense, we have not concentrated our experiments on not using dropout to perform a controlled setting. We have done this because we believe this is currently the relevant scenario regarding mainstream SOTA convolutional networks. We think that strictly batch normalized setting is the one that is most relevant from this point of view because this is the approach followed the SOTA ConvNet models recently.
>
> Firstly, the Inception models are designed without using dropout after convolutional layers. Instead, after those layers, only batch normalization is applied. In those models, dropout was only used before the last fully connected layer.
>
> The original ResNet, an ILSVRC winner, avoids dropout not only after convolutional layers but also before the last fully connected one. The same holds true for the more recent pre-activation ResNet variant. No dropout layers were used. Not even before the densely connected classification layer.
>
> The Wide Residual Network used undoubtedly the same approach. No dropout whatsoever. This network was shown to improve the performance when compared Residual Networks.
>
> The generators and discriminators networks of Generative Adversarial Networks (GANs) are typically convolutional neural networks. Once again, we see no dropout being used in those architectures.
>
> Finally, the DenseNets paper, which won the CVPR 2017 Best Paper Awards, shows in its Table 2 that their strictly batch normalized variants undoubtedly outperforms the options using dropout.
>
> Actually, once established our scope with the above justifications, we indeed performed extremely uncontrolled settings. Different from previously proposed activation functions, we have used standard (no hand-designed) widely used ConvNet models: VGG and ResNets and covered a significant range of depths.
>
> Besides, different from previous works, we execute as many repetitions (runs) of each experiment as needed to achieve statistical significance (p<0.05). We consider this is a significant innovation regarding deep learning papers. Another relevant improvement in methodology was to compare the proposed activation function with many others known non-linearities, not only ReLU.
>
> Regarding the first comment, we showed in Appendix C.6 how the delta was defined. Similarly to LReLU, PReLU, and ELU, in practical usage, the network designer may choose to use 0.05 (DReLU default hyperparameter value) that was established for CIFAR10 if he wants to avoid training costs. Optionally, the designer may perform cross-validation to the specifically used dataset.

---

> > ### Comment · AnonReviewer3 · 2017-12-22
> > **Thanks for the response**
> >
> > Thank you for the response.
> >
> > One additional point: If the improvement can be more significant and on a larger dataset, the work can be much more exciting. The "theory" is straightforward, so the experimental results are what finally matters. I like the idea of statistic test, but still, the improvement of the mean value is limited. The conclusion can be stronger if every model is trained for a sufficiently long time (i.e., converged no matter how many epochs) and the proposed model can beat the state-of-the-art by a significant margin. (This is what I referred by an uncontrolled setting)

---

### Official Review · AnonReviewer1 · 2017-11-28
**Reivew**

**Rating:** 5
**Confidence:** 5

**Review:**

This paper describes DReLU, a shift version of ReLU. DReLU shifts ReLU from (0, 0) to (-\sigma, -\sigma). The author runs a few CIFAR-10/100 experiments with DReLU.

Comments:

1. Using expectation to explain why DReLU works well is not sufficient and convincing. Although DReLU’s expectation is smaller than expectation of ReLU, but it doesn’t explain why DReLU is better than very leaky ReLU, ELU etc.
2. CIFAR-10/100 is a saturated dataset and it is not convincing DReLU will perform will on complex task, such as ImageNet, object detection, etc.
3. In all experiments, ELU/LReLU are worse than ReLU, which is suspicious. I personally have tried ELU/LReLU/RReLU on Inception V3 with Batch Norm, and all are better than ReLU.

Overall, I don’t think this paper meet ICLR’s novelty standard, although the authors present some good numbers, but they are not convincing.

---

> ### Author Response · Authors · 2017-12-02
> **Answers to comments of Reviewer**
>
> Regarding the first comment, we emphasize that the reduction of the mean activations produced by DReLU was not the only argument used to explain why DReLU works better than the other evaluated non-linearities. In fact, theoretical reasons were also provided. Furthermore, it was mentioned and mathematically expressed that DReLU probably implicates less damage to the normalization process as it extends the linear function into the third quadrant, which is a characteristic provided by neither LReLU nor ELU. Moreover, rigorous statistical tests and the vast amount of repetitions of the experiments also contributed from an experimental point of view to ensure DReLU improves the deep models presented.
>
> In respect of the second comment, we argue that CIFAR10/100 are the standard and most frequently used datasets in deep learning computer vision research papers. Moreover, for each dataset, the paper presented consistent results for a range of standard and relevant models with a substantially different number of layers. We did not perform few experiments, rather we executed hundreds of repetitions on the mentioned datasets and performed statistical tests (p<0.05). Naturally, it is never possible to be sure the results presented on some datasets will repeat in other bases. Unfortunately, we will probably not have enough time to execute more 150 experiments needed to include the ImageNet in this study in the next few weeks.
>
> Regarding the third comment, we believe that one of the primary results of our work is precisely showing that without dropout, ReLU is likely to outperform all previously proposed activation functions. It is entirely consistent with the fact that all recently introduced models (VGG, ResNet, WideResNet, DenseNets, etc.) still use ReLU as default activation function. It is no surprise considering the results of our work. Hence, we are not contesting that ELU/LReLU may outperform ReLU if dropout is used. However, our work shows this is unlikely to happen in convolutional networks optimized to strictly use batch normalization without dropout, which is the mainstream state-of-the-art (SOTA) approach to design convolutional networks. These SOTA designs still rely on ReLU as the standard activation function.
>
> Indeed, It is relevant to observe that the mentioned studies usually completely avoid dropout (ResNet, WideResNet, and GANs) or show that the variant without dropout clearly outperforms the one using it (DenseNets). Therefore, we emphasize that no dropout was used since we believe that adding it implies worst results than just use batch normalization, at least for convolutional neural networks. Since 2014 we have seen dropout increasingly less relevant in the design of SOTA ConvNets.
>
> The above mention arguments are indeed in agreement with the more than three hundred experiments and statistical tests we performed which shows that ReLU is a compelling option in strictly batch normalized ConvNets, which is, in our opinion, the best possible design from a regularization point of view to achieve higher performances. Indeed, the test accuracies presented by the paper are essentially state-of-the-art for the models and datasets considered. Besides, dropout slows the training.
>
> We remember that, as mentioned in the paper, the vast majority of the previously proposed activation functions used experiments with dropout and almost always without batch normalization as many of them were designed before the advent of it. We believe that if the experiments with Inception V3 that you mentioned used dropout, it could explain the reason why ELU/LReLU/RReLU outperformed ReLU. If not enough executions were performed or statistical tests were not used, it could also be a statistical error. Finally, different from our study, we emphasize that the previous proposed nonlinearity works did not use standard models (but rather very hand-designed ones), perform statistical tests or at least execute many times the same experiment.
>
> In fact, the use of the statistical tests has been shown to be of fundamental importance as the experiments showed a substantial overlap of the test accuracy performance of the compared activation functions. Therefore, this work showed that make conclusions with few repetitions is inappropriate. Moreover, we made a comprehensive systematic study, testing simultaneously the main activation functions currently in use.

---

### Decision · Program_Chairs · 2018-01-29
**ICLR 2018 Conference Acceptance Decision**

**Decision:**

Reject

**Comment:**

meta score: 4

This paper proposes an activation function, called displaced ReLU (DReLU), to improve the performance of CNNs that use batch normalization.

Pros
 - good set of experiments using CIFAR, with good results
 - attempt to explain the approach using expectations
Cons
 - theoretical explanations are not so convincing
 - limited novelty
 - CIFAR is relatively limited set of experiments
 - does not compare with using bn after relu, which is now well-studied and seems to address the motivation of this paper (and thus questions the conclusions)